# Utilizing Tunnel Boring Machine (TBM)-Crushed Limestone as a Construction Material

**DOI:** 10.3390/ma15217569

**Published:** 2022-10-28

**Authors:** Ahmed Alnuaim, Ahmed M. Al-Mahbashi, Muawia Dafalla

**Affiliations:** Department of Civil Engineering, College of Engineering, King Saud University, Riyadh, P.O. Box 800, Riyadh 11421, Saudi Arabia

**Keywords:** crushed limestone, resilient modulus, pavement, tunnel boring machine

## Abstract

Tunnel boring machine (TBM) materials are usually crushed powder obtained during tunnel excavations for subways and transportation networks. Huge quantities of crushed rock powder are generally treated as waste. This study is aimed at assessing proposed mixtures of TBM and granular material for use in construction. This approach will help in a greener environment and reduce the need for crushed aggregates used in sub-base and subgrade layers of pavements. Assessment is executed using dynamic and static strength tests, including the modulus of resilience and the California bearing ratio (CBR). The TBM-crushed material can be sorted and screened on site to optimize its use as a construction material. The blending ratios for the 3/8-inch aggregate (G1) to the material-passing sieve number 4 (P4) were found to influence the pavement design parameters. This study recommends sorting the TBM-crushed limestone by an on-site sieving operation. A guide to optimizing the quality of the material is suggested by blending 3/8-inch aggregate with the crushed limestone fine-powder material at a specified percentage ranging from 5 to 10% by weight. The stability and durability tests conducted on the TBM-crushed powder material confirmed its suitability as a pavement construction material for subgrade and sub-base layers. Modulus of resilience, CBR values and compressive strength tests were carried out for different suggested mixtures.

## 1. Introduction and Objectives

In the Riyadh metro project of Saudi Arabia, massive heaps of tunnel boring machine (TBM)-crushed limestone powder and chips of limestone were stockpiled as a waste material. The concept of a greener environment and recycling are developed to investigate the reuse of this product in various applications. Tunneling debris obtained from excavations using tunnel boring machines is currently a great concern and challenge. It needs to be utilized as an environmentally friendly product in construction works that include concrete production, pavement applications and other needs [1]. The use of tunnel chips as a concrete constituent has been studied by a few researchers [2,3,4]. Berdal [5] observed that the TBM systems could produce a variety of spoil sizes depending on the machine used. It was suggested that the TBM spoil be utilized as a concrete aggregate where it is economically beneficial. The study showed concerns regarding the high fine content when used in concrete mix designs.

The works of Riviera et al. [6] investigated the reuse of tunnel waste products in the subgrade and sub-base layers for roads and pavements, as well as embankments. The waste tunnel material may be rejected for use in a high-quality base course or other specific layers due to specifications set by AASHTO M145 (2003) [7] or similar agencies. This could be due to oversize, flakiness, elongated shape or mismatch of other requirements.

Alnuaim et al. [8] studied the utilization of the crushed limestone powder obtained from the TBM operations as an additive to liners for the containment of domestic waste dumping sites. It was found that the addition of TBM-crushed limestone powder can improve the compressibility of clay–sand liners and reduce the swell index when one third or two thirds of bentonite are replaced by this material. Moreover, the dry density of the clay–sand mixture increases when TBM-crushed limestone powder is added, because the powder occupies voids within the bentonite clay or fine sand mixtures.

The TBM material properties depend on the type of rock in the area where tunnels are excavated. Riyadh is generally underlain by limestone formations. The TBM powder obtained throughout Riyadh is mostly of similar nature and composition, except for the size and heterogeneity.

The Saudi Geological Society [9] described the Riyadh area as being underlain by Phanerozoic sedimentary formation resting on the Precambrian rocks of the Arabian Shield. Late tertiary rocks overlie the Phanerozoic rocks. Quaternary deposits of silt, sand and gravel with variable cementations are dominant. The subsurface material below Riyadh metro was found to include light-brown, fine- to medium-grained calcarenite with layers and inter-beds of siliceous limestone. 

The quality of material used in the subgrade and sub-base layers is vital for a durable and stable pavement. Many local authorities require specific characteristics for pavement construction. This is normally based on their local needs and the available resources. These materials vary in their lifetime and performance. The use of alternative materials to enhance subgrade stability is a worldwide practice to achieve quality materials that satisfy specifications and standards. Riyadh municipality generally calls for A-1-a and A-1-b material (and occasionally A-2-4) for the subgrade and sub-base layers (AASHTO M 145) [7].

Meeting the classification requirements can be achieved by sorting and mixing the crushed limestone product of the tunnel boring machine (TBM) in a certain order. The major challenge for soils with dominant fines will be the stability requirements. These are commonly assessed using the Modulus of Resilience tests (MR) or the California Bearing tests (CBR). Other evaluation tests used in the field assessment include the dynamic cone penetrometer (DCP) and the static cone penetrometer (SCP). The Illinois Department of Transportation (IDOT) has successfully used the dynamic cone penetrometer (DCP) and the static cone penetrometer (SCP) tests to determine the subgrade stability in a field mining subgrade evaluation.

This study was conducted to evaluate the use of TBM as a natural material for designing the subgrade layers of pavement projects. Three mixtures with different gradations were proposed, and examined for suitability and compliance with recognized standard requirements. In addition to the basic properties, strength and quality tests have been completed, including resilient modulus (M_r_), Los Angeles California bearing ratio (CBR) and unconfined compression strength (UCS). In light of these results, the suitability of these mixtures as successful subgrade materials is discussed and highlighted. 

## 2. Materials and Methods

### 2.1. Materials

The TBM spoil material is a crushed powder of limestone. It is generally a poorly graded material, as classified using the unified soil classification system (USCS) or the ASTM D 2487 [10]. Table 1 presents the characterization test results performed in the laboratory. The specific gravity was 2.714, which was a little higher than that measured for the quaternary sand deposits. The plasticity index of the crushed powder was very low and reported as low as 3. The optimum moisture content was reported as 16 at a maximum dry density of 17.6 kN/m^3^.

### 2.2. Testing Methods

#### 2.2.1. Sieve Analysis and Classification Tests

The material imported from the stockpiles stored nearby the tunnels was of variable texture and size. It was sorted in the lab by simple screening using sieves of selected openings. The material included fines, extra fines, coarse and rock fragments. The sorting was completed in order to utilize the TBM-crushed limestone powder as a road or pavement material. This can be enabled by sorting and re-mixing to achieve acceptable subgrade, sub-base and pavement materials. The excavated material brought to the laboratory was tested for grain size distribution following the ASTM D6913 [11] requirements. The grain size distribution curves are presented in Figure 1. Plasticity tests (ASTM D 4318) [12] for the excavated natural material were conducted and used for the purpose of characterization and classification as per the ASTM D 2487 [7]. The sorted material was blended in different proportions to investigate its suitability for use as an AASHTO-approved road material. Blends of 3/8-inch aggregates (G1) were added to the TBM material passing through sieve number 4 (P4) at percentages of 5%, 10% and 15% for the determination of the modulus of resilience and the CBR values. Blends including 3/4-inch (P3/4) aggregate were found not to satisfy the gradation ranges established for A-1-a and/or A-1-b.

#### 2.2.2. Dry Density Moisture Content Relationship

Standard compaction characteristics were carried out in accordance with ASTM D 698 [13]. Figure 2 presents the compaction characteristics of the TBM-produced soil and other considered mixtures. For the purpose of conducting the CBR test another compaction test specification was used (ASTM D 1557) [14].

#### 2.2.3. Soil Water Characteristic Curves

The soil water characteristic curves (SWCC), expressed as gravimetric water content versus matric suction, were constructed for the compacted tunnel boring machine (TBM) powder material.

The pressure plate method, or as it is also called the axis translation technique, was performed for a wide range of suction pressures. Figure 3 presents the SWCC of the mixtures using the TBM powder. The test was performed in accordance with the standard procedure described in ASTM D6836 (2008) [15]. The test data were fitted using the Fredlund and Xing [16] fitting equation, as shown in Equation (1) below:(1)$$w\left(\Psi \right)=w_{s}\left[1-\frac{\ln\left(1+\frac{\Psi }{h_{r}}\right)}{\ln\left(1+\frac{10^{6}}{h_{r}}\right)}\right]\left[\frac{1}{ln{\left\{\exp\left(1\right)+{\left(\frac{\Psi }{a}\right)}^{n}\right\}}^{m}}\right]$$
where 

*Ψ* = soil suction; 

*w* = gravimetric water content; 

*w_s_* = saturated gravimetric water content; 

*w_r_* = residual gravimetric water content; 

*a* = suction related to the inflection point on the curve; 

*n* = soil parameter related to slope at the inflection point;

*m* = soil parameter related to the residual water content; 

*h_r_* = suction related to the gravimetric residual water content.

**Figure 3 materials-15-07569-f003:**
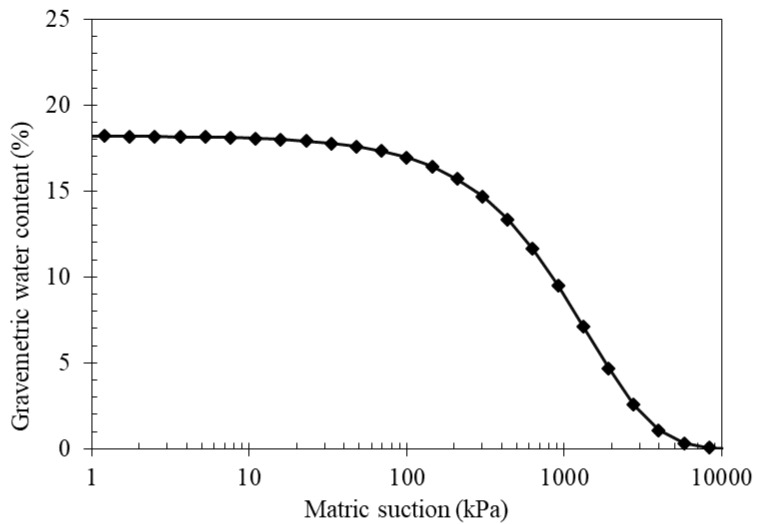
Soil water characteristic curve (SWCC) of the TBM soil material.

#### 2.2.4. Modulus of Resilience Tests

The resilient modulus tests were performed in a triaxial dynamic loading system; the device consists of a load frame with a 50 kN electro-mechanical actuator controlled by a dynamic servo controller and pressure cell provided with load cell and displacement transducers, as shown in Figure 4.

The tests were carried out following the method described in AASHTO T307-99 [17] for compacted cylindrical specimens 50 mm in diameter and 100 mm high. The specimen was encased by a rubber membrane and mounted into the device cell. Three confining pressures (σ_c_) were considered, and at each confining pressure five maximum deviator stresses (σ_d_) were reached. The corresponding strains were monitored through attached transducers; the resilient modulus was calculated as a ratio between the changes in the stress to the corresponding strain change. More details about test procedures are available in Elkady et al. [18]. M_R_ tests were performed for natural TBM material and the designed mixtures with percentages of 5%, 10% and 15% of aggregate G1. The term G1 refers hereafter to the size of the aggregates passing through the 4.75 mm sieve and retained on the 10 mm sieve.

**Figure 4 materials-15-07569-f004:**
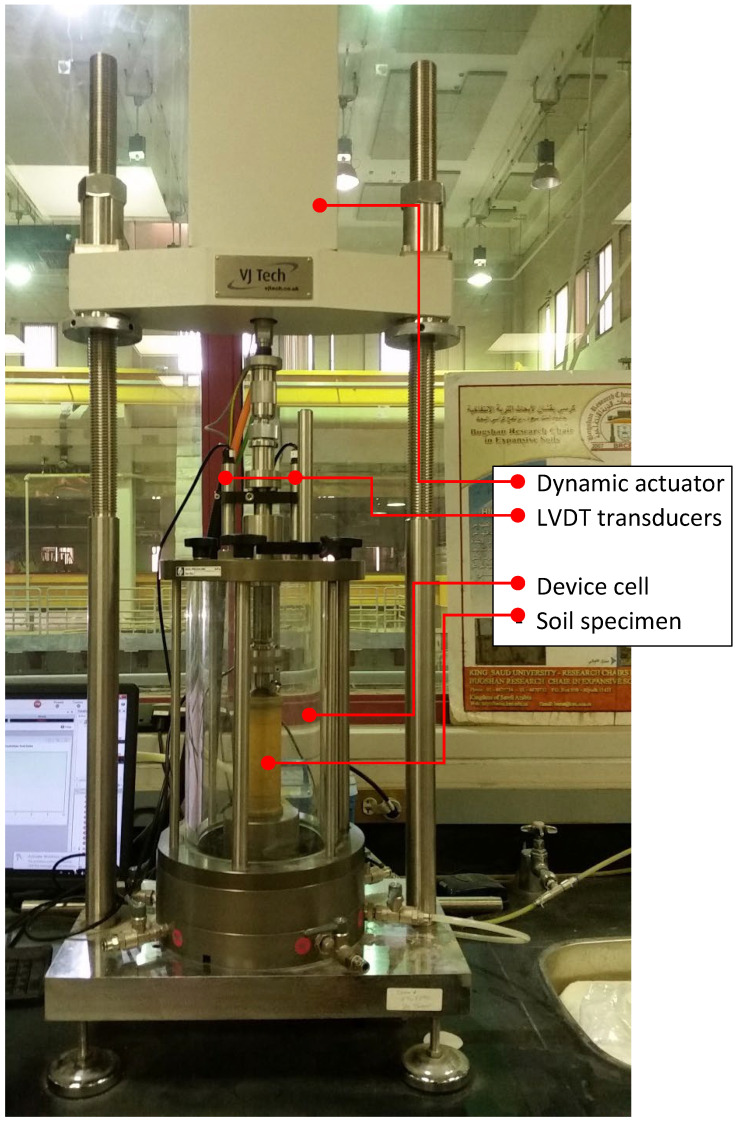
Resilient Modulus device.

#### 2.2.5. Compressive Strength

The mixtures used for the modulus of resilience tests were also tested for the unconfined compressive strength in a universal frame equipped with a hydraulic loading system and a load cell.

#### 2.2.6. California Bearing Ratio (CBR) Tests

The California bearing ratio (CBR) test is one of the main stability tests performed to assess the strength and stiffness of the materials used for the base, sub-base, subgrade and embankment layers in pavements. This is a load deformation test conducted in laboratories or in the field. The CBR test was conducted in accordance with ASTM D1883-14 [19] for the selected blends of aggregates. Samples were prepared by mixing the blended dry material and by compaction of the mixtures to the optimum moisture content (OMC) and the maximum dry density (MDD) of the blended mix.

#### 2.2.7. Resistance to Abrasion

In order to assess the quality of the material with regard to abrasion, the AASHTO T 96/ASTM C 131 [20,21] test was conducted. This test investigates the resistance to the degradation of aggregates by abrasion and impact in the Los Angeles machine. Table 2 presents Los Angeles abrasion test result.

#### 2.2.8. X-ray Diffraction and SEM

The X-ray diffraction for the TBM powder material was conducted with radiation produced from a Cu target at a voltage of 40 kV and a current of 30 mA. Fabric features of the TBM material were viewed using EDAX in a scanning electron microscope apparatus (Joel Model JSM-7600F with a high resolution of 3.0 nm and operated at 5–10 kV). This technology can provide good visibility and is considered more advanced. All tests were carried out on specimens compacted to the maximum dry density corresponding to the optimum water content. A freeze-drying technique using liquid nitrogen was performed for the compacted specimens in order to avoid any disturbance in the fabric of the tested specimens.

## 3. Results

The classification test results presented in Section 2 above show the nature of the crushed limestone TBM material. The compressibility and soil water characteristics curve define the maximum dry density, optimum moisture content and soil suction at varying moisture content. 

The unit used for testing the resilient modulus was employed to produce three different confining pressures: 13.8, 27.6 and 41.4 kPa. The maximum axial strength was reported for each mixture.

The compressive strength for the TBM materials and mixtures indicated an initial elastic behavior before reaching a maximum strength. A very brief transition zone was observed prior to failure, followed by large strain and reduced stress values.

The applied stress versus penetration values for mixtures, including 5% and 10% aggregates, are tested. The trend shows a linear relationship, indicating an increase in penetration with the increase in stress. 

## 4. Discussion

The traditional excavation methods for tunnels using conventional types of machinery and hammers produce heterogeneous waste materials, including boulders and fragments of different shapes and sizes. Modern technology introduced tunnel boring machines capable of producing materials with limited variation in size and shape. This is actually dependent on the cutting tools utilized. Diamond or carbide inserts fitted into the cutting bits can produce fine powder or medium to coarse size cuttings. This study is aimed at studying the characterization of crushed limestone for possible use in pavement construction. The TBM-crushed limestone obtained from the Riyadh metro was found to be of very low plasticity and low clay content. Table 1 presents the characterization summary of the material generally referred to as TBM-crushed limestone material.

The moisture–dry density relationship is an important criterion that helps in obtaining better compaction with less effort, provided that the mixtures are prepared at the optimum moisture content. The water intake is high when the material involves more clays and fines [22]. The soil water characteristic curve presented in Figure 3 provides the relationship between the gravimetric moisture content and suction for the TBM material used in this study. The soil water characteristic curve indicated low suction (nearly 100 kPa) at a moisture content of 16%. The optimum moisture content for the TBM material was found in the order of 16 percent at 17.6 kN/m^3^ maximum dry density.

Initial trials for blending the TBM material (70%) with construction sand (10%), 3/4-inch (10%) and 3/8- or G1- (10%) inch aggregates yielded a material with significant gaps in gradation. This is considered to be a poor rating for use in pavement construction. Sieves (1-inch, 3/4-inch and 3/8-inch) were all found out of range and cannot match A-1-a or A-1-b in the AASHTO classification. This material yielded high CBR (in excess of 100%) values due to the presence of coarse aggregate, but cannot be recommended for use in the pavement. All the blends, including 3/4-inch aggregates, appear to be not appropriate due to the gap in the gradation. This is also occurring with the smaller aggregate 3/8-inch size (G1), but the gap is smaller and the mixture can be classified within acceptable groups of AASHTO M 145.

The Resilient modulus tests for the TBM materials and the three suggested blends of 5%, 10% and 15% aggregates (3/8-inch aggregate, G1) added to the TBM powder were conducted. The confining pressures of 13.8, 27.6 and 41.4 kPa were adopted for five different maximum axial dynamic loads. Figure 5, Figure 6 and Figure 7 present the test results.

It can be clearly observed that the MR is reduced by adding more aggregates. The addition of 5–10% aggregates was found to give values comparable to that of the TBM material without addition. The reduction in the MR is not significant when the added material is less than 10%. This can be explained due to the voids within the aggregate not being filled up with the TBM powder material. The modulus of resilience is reported in the range of 350 to 600 for the TBM material blended with less than 10% aggregate at the maximum dynamic loads of 15 kPa, and in the range of 200 to 250 for the TBM material blended with less than 10% at the maximum dynamic loads of 70 kPa. The addition of 15% G1 aggregate indicated a very low modulus of resilience (100 to 150) for the load intensities investigated.

Figure 8 presents the unconfined compressive strength of TBM material and mixtures including 5%, 10% and 15% aggregates. The strength of 5% and 15% aggregate mixtures reported higher strength compared to the 10% additive. The high strength is likely attributed to the well-graded nature of the 5% mix and the excessive aggregates in the case of the 15% mix. Figure 9 shows a deeper extending shear crack for the 10% mix compared to the other two mixtures.

California bearing ratio tests were conducted for the blended mixtures; the mixture with 5% aggregate yielded a CBR value of 42, while the mixture with 10% aggregate yielded a CBR value of 47 (Figure 10). The blended mixture with 15% aggregate was excluded and not tested for the CBR due to the low modulus of resilience reported in the dynamic loading test. 

The X-ray diffraction results showed a composition similar to what was seen for minerals in the Riyadh limestone. Calcite (CaCO_3_) is the dominant mineral present. The dolomite is also present in the Riyadh area but not shown in the selected samples. Quartz (SiO2) appeared in many clear peaks. Clay minerals peaks are likely influenced by the background intensities. Aluminum oxide peaks can be detected but the intensity background within the low angles is higher than the average background. Figure 11 depicts the peaks obtained from the X-ray diffraction of the TBM powder material over a selected range of (two theta). It can be deduced from the intensity plots that the calcium carbonate content is in the order of 20%, which is in good agreement with the EDS results obtained for the crushed limestone powder. No pozzolanic compounds were seen. The main common constituents included quartz, calcite, siderite and alumina; the information provided is only qualitative. Minor peaks within the clay minerals zone are likely influencing the background intensity for (two theta) above 20 degrees.

From the scanning electron microscope (SEM) micrograph, the qualitative and quantitative printouts of the EDS results are presented in Figure 12. TBM material indicates crushed fines occupying most of the pores and voids.

Most of the studies reviewed in this research highlighted the re-use of TBM materials in concrete applications. The works of Riviera et al. [6] suggested stabilization approaches for roads and pavements. This study introduces the concept of sorting and mixing of the waste material as a cheaper approach. Mitigation methods suggested by reference [6] involved chemical stabilization and possibly expensive additives. The current study also provides quantitative solutions which can serve as a guide for trials to satisfy the requirements and material specifications of different pavement components. 

## 5. Conclusions

Tunnel boring machines (TBM)-crushed powder product can be confidently used for roads and pavement construction provided that sorting is performed for at least three grades. These include the material passing the 3/8-inch sieve, the material passing the 3/4-inch sieve, and retained on the 3/8-inch sieve in addition to the material retained on the 3/4-inch sieve. This study indicates that the crushed limestone produced by the TBM consists of dominant fine-grained fragments, which can turn into a poorly graded or a gap-graded mix if aggregates are added to improve stability and strength. Based on the dynamic modulus of resilience tests conducted, it is concluded that 3/8-inch aggregate (G1) can be utilized and added to the material, without compromising the stability and strength requirements of the subgrade and sub-base layers of pavements. Addition of excessive material (over 10%) can drastically reduce the MR of the crushed limestone powder. Large-size aggregates of limestone are not suitable as additives. The presence of oversize fragments is a reason for material rejection due to its influence in reducing compaction and forming voids within the layers.

The soil water characteristic curves indicated low suction values for the ranges close to the optimum content. The stability and durability tests conducted on the TBM-crushed powder material confirmed its suitability as a pavement construction material for subgrade and sub-base layers. This is also valid for general embankment purposes. The Los Angles Abrasion test of 25 was measured. CBR values of 42 and 47 were obtained for TBM material mixed with 5% and 10% aggregate G1 (3/8-inch). This is considered satisfactory for the types of layers used in practice. Compressive strength indicated that 5% aggregate is an optimum mixture to provide a stable subgrade material. The main common constituents seen in the microstructure analysis included quartz, calcite, siderite and alumina.

## Figures and Tables

**Figure 1 materials-15-07569-f001:**
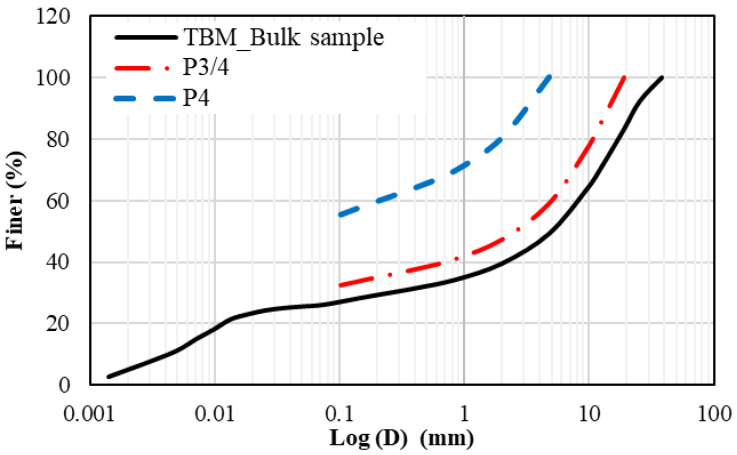
The grain size distributions for sorted TBM material (spoil) crushed limestone material.

**Figure 2 materials-15-07569-f002:**
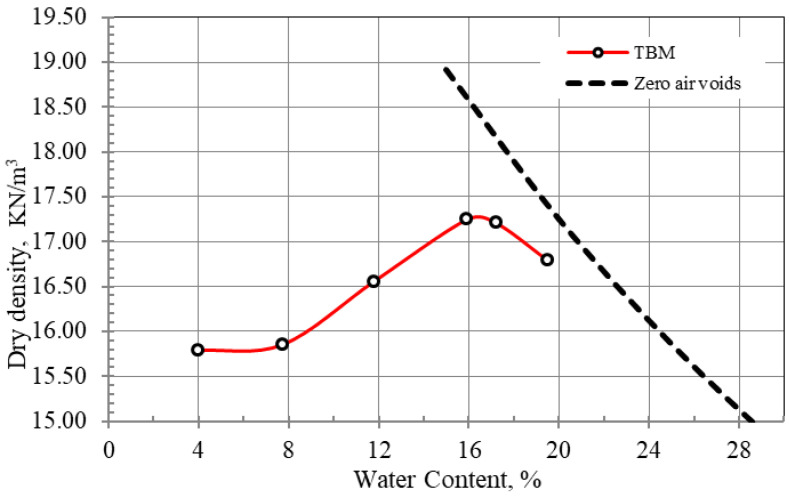
Compaction characteristics of TBM soil.

**Figure 5 materials-15-07569-f005:**
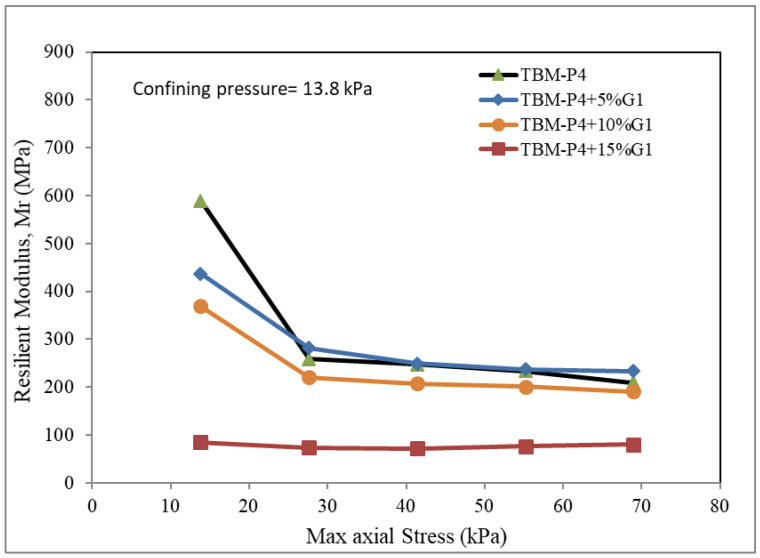
Resilient modulus under confining pressure 13.8 kPa.

**Figure 6 materials-15-07569-f006:**
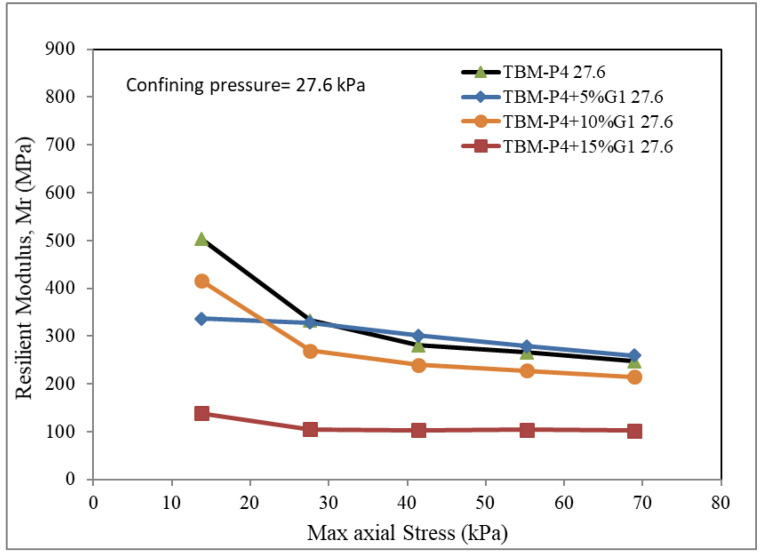
Resilient modulus under confining pressure 27.6 kPa.

**Figure 7 materials-15-07569-f007:**
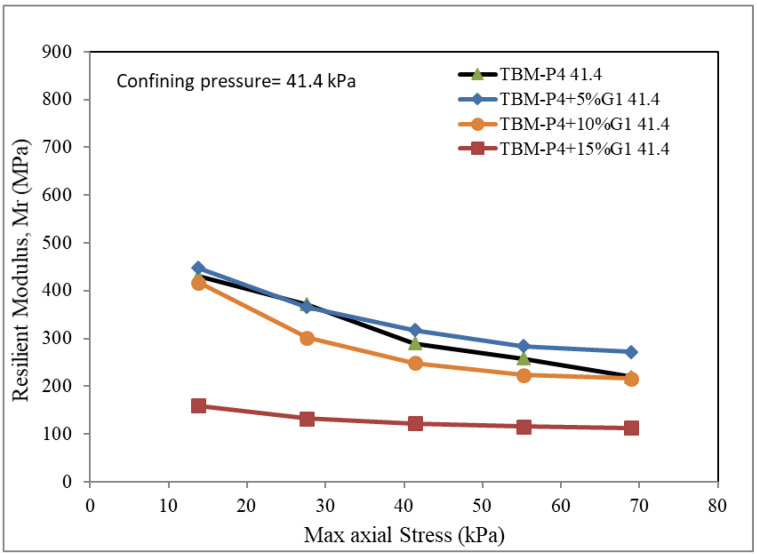
Resilient modulus under confining pressure 41.4 kPa.

**Figure 8 materials-15-07569-f008:**
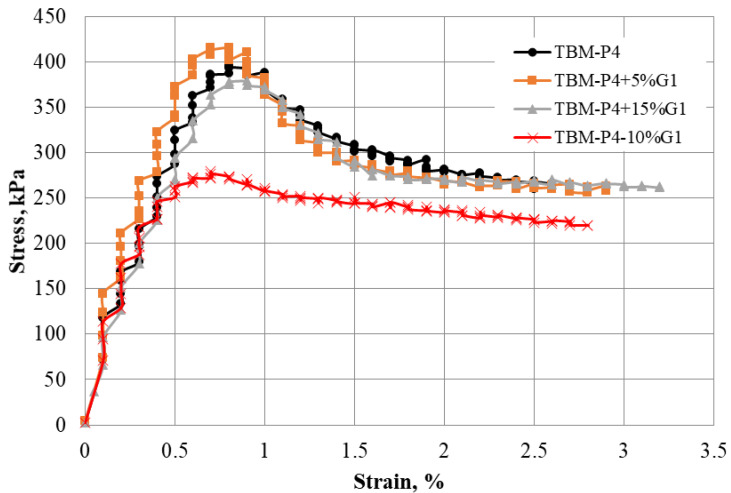
Unconfined compressive strength for the TBM material and mixtures with aggregate.

**Figure 9 materials-15-07569-f009:**
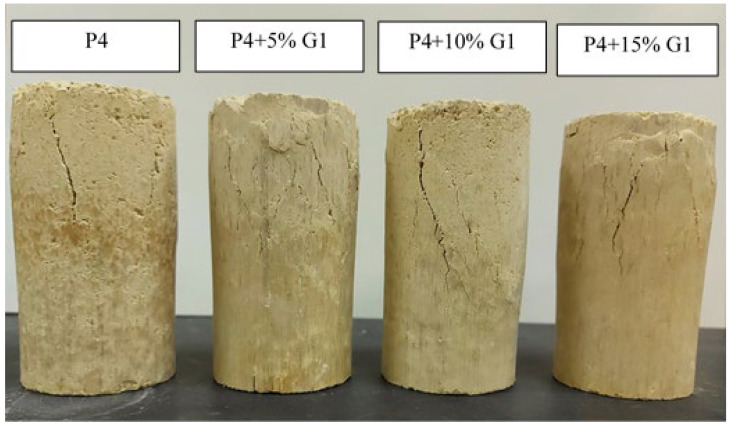
Modes of failure under compression of TMB material and mixtures with 5%, 10% and 15% aggregate.

**Figure 10 materials-15-07569-f010:**
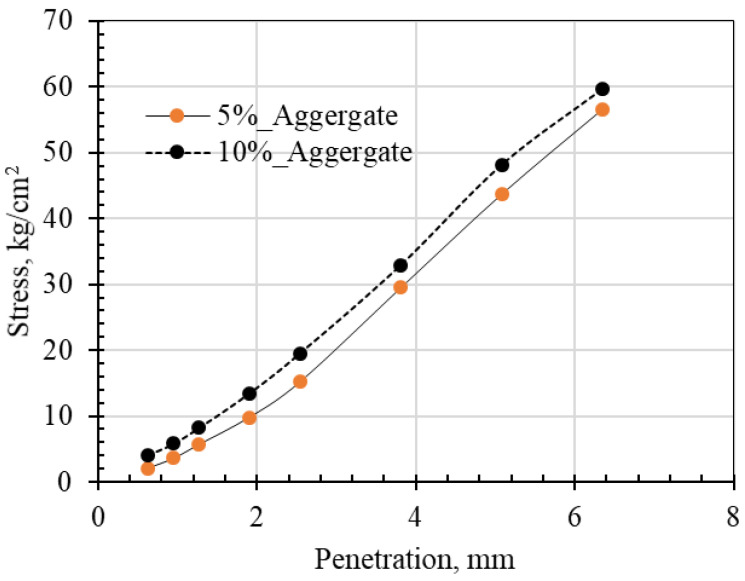
Penetration versus unit load for CBR tests.

**Figure 11 materials-15-07569-f011:**
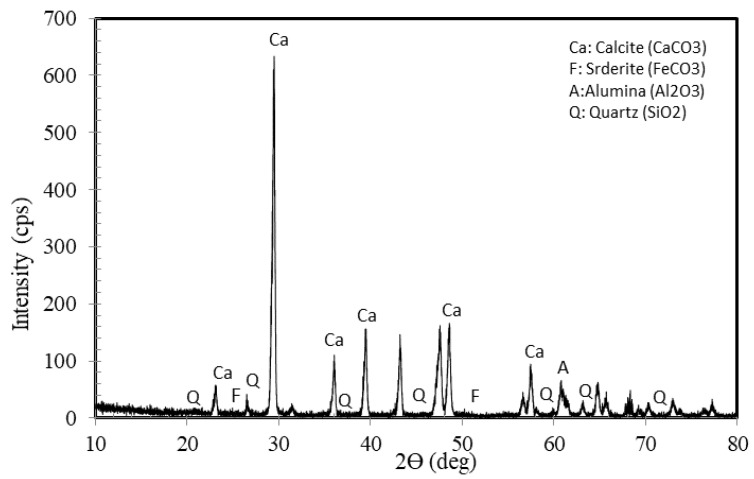
XRD profile for TBM powder material.

**Figure 12 materials-15-07569-f012:**
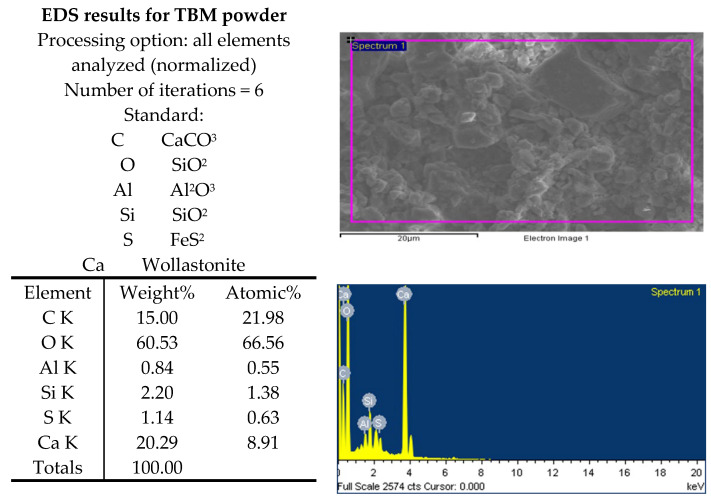
EDS results for TBM powder.

**Table 1 materials-15-07569-t001:** Characterization of TBM soil.

Test	Value
Specific Gravity	2.714
Liquid Limit (%)	22
Plastic Limit (%)	19
Shrinkage Limit (%)	18
Plasticity Index (%)	3
Compaction characteristics (ASTM D 698-2000)	
Optimum moisture content (%)	16.5
Maximum dry unit weight (KN/m^3^)	17.6

**Table 2 materials-15-07569-t002:** Los Angeles Abrasion Tests.

Description	Results
No. of spheres	8
Total weight of test sample	5000
Weight of tested sample retained on sieve no. 12	3765.8
Weight of tested sample passing through sieve no. 12	1234.2
% Loss after 500 revolutions	24.7
Specification limit: % maximum	45
Status	Pass

## Data Availability

The data used to support the findings of this study are included in the introduced figures.

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
