# Peer review of "Utilizing Tunnel Boring Machine (TBM)-Crushed Limestone as a Construction Material"

_materials, 2022, doi:10.3390/ma15217569_

Round 1
Reviewer 1 Report
This study evaluated the properties of tunnel boring machine materials and granular material mixtures for their potential use as construction materials. The dynamic and static strength tests were performed to evaluate the properties of the TBM and an optimized mixing ratio of the aggregate was recommended according to the test results. However, the manuscript was not well organized and some parts of the paper were unclear. There are some issues that need to be addressed:
1. In Figure 1, what are natural samples, natural soil, and sampling standard for? Which one is the TBM material?
2. The results of the test were only the list of the figures, the description of the results was needed for the results part.
3. The discussion of the results was not deep enough, it’s more like a report instead of a research paper.
4. The novelty of this study was not clear, most of the tests conducted in this study were the standard tests. In addition, it is unclear if the findings of this study were suitable for TBM materials from other sources.
Reviewer 2 Report
Paper ID:
materials-1971991
Type:Article
Title: Utilizing Tunnel Boring Machine (TBM) Crushed Limestone as a Construction Material
Authors: Ahmed Alnuaim, Ahmed Al-Mahabashi, Muawia Dafalla
This paper assesses proposed mixtures of TBM and granular material for use in construction. Following are some comments for the authors to consider:
1. Novelty in comparison to recent literature? Need to be emphasized.
- The results in the paper might be more discussed by the relevant literature.
- You might briefly state method of this study in the last paragraph of Introduction section.
- Throughout the text, there are some typos that must be eliminated.
- In the study, the use of abbreviations was not paid attention in general. All abbreviations need to be reviewed.
- Line 126: Figure 4 ? or Figure 3 ?
- Attention should be paid to the spelling of subscripts and superscripts.
Round 2
Reviewer 1 Report
The authors have addressed part of the comments, however, some issues still need to be discussed. In the introduction part, the authors presented some studies that used tunnel waste for the subgrade and sub-base layer for roads and pavements, how the findings of this study are different from the existing study? The new findings of this study should be emphasized in the manuscript.
Author Response
Response to reviewer comment:
Thank you for your constructive comments. An additional paragraph is added at the end of the discussion section to highlight the difference between the current study and previous works.
_________ Added text: _____
" Most of the studies reviewed in this research highlighted the re-use of TBM materials in concrete applications. The works of Riviera et al. [6] suggested stabilization approaches for roads and pavements. This study introduced the concept of sorting and mixing of waste material as a cheaper approach. Mitigation methods suggested by reference [6] involved chemical stabilization and possibly expensive additives. The current study also provided quantitative solutions which can serve as a guide for trials to satisfy the requirements and material specifications of different pavement components. "
Reviewer 2 Report
Authors have made necessary changes. The paper can therefore be accepted.
Author Response
Thank You.